# Co-resistance to isoniazid and second-line anti-tuberculosis drugs in isoniazid-resistant tuberculosis at a tertiary care hospital in Thailand

Ajala Prommi,[1,2] Kanphai Wongjarit,[3,4] Suthidee Petsong,[3] Ubonwan Somsukpiroh,[5] Kiatichai Faksri,[6,7] Kamon Kawkitinarong,[8,9] Sunchai Payungporn,[2,10] Suwatchareeporn Rotcheewaphan[2,3]

**ABSTRACT**   Isoniazid-resistant tuberculosis (Hr-TB) is an important drug-resistant tuberculosis (TB). In addition to rifampicin, resistance to other medications for Hr-TB can impact the course of treatment; however, there are currently limited data in the literature. In this study, the drug susceptibility profiles of Hr-TB treatment and resistance-conferring mutations were investigated for Hr-TB clinical isolates from Thailand. Phenotypic drug susceptibility testing (pDST) and genotypic drug susceptibility testing (gDST) were retrospectively and prospectively investigated using the Mycobacterium Growth Indicator Tube (MGIT), the broth microdilution (BMD) method, and whole-genome sequencing (WGS)-based gDST. The prevalence of Hr-TB cases was 11.2% among patients with TB. Most Hr-TB cases (89.5%) were newly diagnosed patients with TB. In the pDST analysis, approximately 55.6% (60/108) of the tested Hr-TB clinical isolates exhibited high-level isoniazid resistance. In addition, the Hr-TB clinical isolates presented co-resistance to ethambutol (3/161, 1.9%), levofloxacin (2/96, 2.1%), and pyrazinamide (24/118, 20.3%). In 56 Hr-TB clinical isolates, WGS-based gDST predicted resistance to isoniazid [*katG* S315T (48.2%) and *fabG1* c-15t (26.8%)], rifampicin [*rpoB* L430P and *rpoB* L452P (5.4%)], and fluoroquinolones [*gyrA* D94G (1.8%)], but no mutation for ethambutol was detected. The categorical agreement for the detection of resistance to isoniazid, rifampicin, ethambutol, and levofloxacin between WGS-based gDST and the MGIT or the BMD method ranged from 80.4% to 98.2% or 82.1% to 100%, respectively. pDST and gDST demonstrated a low co-resistance rate between isoniazid and second-line TB drugs in Hr-TB clinical isolates.

**IMPORTANCE**   The prevalence of isoniazid-resistant tuberculosis (Hr-TB) is the highest among other types of drug-resistant tuberculosis. Currently, the World Health Organization (WHO) guidelines recommend the treatment of Hr-TB with rifampicin, ethambutol, pyrazinamide, and levofloxacin for 6 months. The susceptibility profiles of Hr-TB clinical isolates, especially when they are co-resistant to second-line drugs, are critical in the selection of the appropriate treatment regimen to prevent treatment failure. This study highlights the susceptibility profiles of the WHO-recommended treatment regimen in Hr-TB clinical isolates from a tertiary care hospital in Thailand and the concordance and importance of using the phenotypic drug susceptibility testing or genotypic drug susceptibility testing for accurate and comprehensive interpretation of results.

**KEYWORDS**   *Mycobacterium tuberculosis*, isoniazid-resistant tuberculosis, drug susceptibility profile, whole-genome sequencing

Tuberculosis (TB) is an important public health problem in Thailand; in 2022, the World Health Organization (WHO) listed Thailand among 30 countries with a high TB

Address correspondence to Suwatchareeporn Rotcheewaphan, suwatchareepon.r@gmail.com.

The authors declare no conflict of interest.

See the funding table on p. 10.

burden (1). Drug-resistant TB (DR-TB) is an emerging global concern, posing challenges to TB treatment and infection control. Multidrug-resistant tuberculosis (MDR-TB) is a *Mycobacterium tuberculosis* (MTB) strain that is resistant to rifampicin (RIF) and isoniazid (INH), while isoniazid-resistant TB (Hr-TB) refers to the MTB strain that is susceptible to RIF but resistant to INH (1, 2). INH is an active drug with bactericidal activity and is used to treat active TB and prevent latent TB reactivation (3). The prodrug INH requires activation by the mycobacterial catalase–peroxidase enzyme (KatG) to interact with enoyl-acyl carrier protein reductase (InhA) and inhibit mycolic acid synthesis in the cell wall. Mutations in the *katG* gene are commonly involved in resistance to INH, followed by the promoter region of the *fabG1-inhA* operon and the *ahpC* promoter (4, 5).

Hr-TB can be diagnosed using conventional mycobacterial culture followed by phenotypic drug susceptibility testing (pDST), such as the agar proportion method, fluorescence-based liquid medium system, or the broth microdilution (BMD) method (6, 7). However, these tests are time-consuming, which can delay the initiation of the appropriate treatment. Therefore, rapid molecular tests are important for the diagnosis of DR-TB. Furthermore, the resistance of MTB to INH can be classified as low- or high-level resistance based on minimal inhibitory concentration (MIC) results when tested on liquid medium: low-level resistance as an MIC of INH greater than 0.1 mg/L to 0.4 mg/L and high-level resistance as an MIC greater than 0.4 mg/L (8). This information plays a critical role in the selection of the appropriate treatment regimen, such as the inclusion of high-dose INH or the exclusion of INH. The current treatment regimen for Hr-TB recommended by the WHO includes RIF, ethambutol (EMB), pyrazinamide (PZA), and levofloxacin (LFX) for a duration of 6 months (3). Therefore, accurate pDST or genotypic drug susceptibility testing (gDST) for INH and other drugs is a key factor in carefully selecting patients who can benefit from the Hr-TB regimen at the start of TB treatment. For gDST, several WHO-recommended molecular tests, namely, Xpert MTB/RIF, Xpert MTB/RIF Ultra, and Truenat MTB-RIF Dx (9), cannot simultaneously detect resistance to INH, RIF, and other anti-TB drugs in the same reaction. Whole-genome sequencing (WGS) is a reliable tool for predicting resistance to first- and second-line drugs of the *M. tuberculosis* complex (MTBC) (5, 10).

Currently, the susceptibility profiles of Hr-TB clinical isolates circulating in Thailand are still limited, especially for drugs in the WHO-recommended treatment regimen. Therefore, this study aimed to address this research gap and evaluate the concordance of susceptibility results from the Mycobacterium Growth Indicator Tube (MGIT) and BMD methods and the WGS approach for the diagnosis of Hr-TB and resistance to the treatment regimen.

## MATERIALS AND METHODS

### MTBC clinical isolates and patient populations

A retrospective study investigated Hr-TB clinical isolates recovered from clinical specimens at King Chulalongkorn Memorial Hospital (KCMH) from January 2017 to December 2021. A total of 183 Hr-TB clinical isolates were recovered from 144 patients with TB based on the pDST results for streptomycin (STM), INH, RIF, and EMB (SIRE) tested by the BD BACTEC Mycobacterium Growth Indicator Tube 960 system (MGIT; BD Biosciences, Franklin Lakes, NJ, USA) in a routine laboratory service without additional clinical specimens requested for this study. The MGIT results of 161 nonduplicate Hr-TB clinical isolates from 144 patients and characteristics of the patients with Hr-TB (age at the time of the first diagnosis of Hr-TB, sex, and treatment history) were recovered from medical records. This study was approved by the Institutional Review Board of the Faculty of Medicine, Chulalongkorn University, Bangkok, Thailand (COA.1297/2019).

### Isolation and identification of the MTBC

The MTBC was isolated from clinical specimens using the sodium hydroxide-N-acetyl-L-cysteine-sodium citrate method, as previously described (11), and frozen at −80°C

until use. The isolates were previously identified using line probe assays (LPAs) such as GenoType Mycobacterium CM VER 2.0 (MTBC level) and GenoType MTBC VER 1. X (MTB species level) (Hain Lifescience, Nehren, Germany) following the manufacturer's instructions. The frozen stock of the mycobacteria was subcultured in Lowenstein–Jensen medium at 37°C for 3 to 4 weeks or until a visible colony was observed and was sufficient for prospective pDST and genomic DNA extraction for WGS analysis.

## Phenotypic drug susceptibility tests

The MGIT 960 SIRE kit was previously tested with the following final drug concentrations (mg/L): STM (1.0), INH (0.1), RIF (1.0), and EMB (5.0), following the manufacturer's instructions. A prospective pDST study was conducted for nonduplicate Hr-TB isolated from an individual patient based on distinguishable SIRE MGIT results or from different types of specimens in each disease episode. Some clinical isolates were excluded from prospective pDSTs due to bacterial contamination and no growth after subculture. Finally, for the MGIT system, the Hr-TB clinical isolates were prospectively tested with high-dose INH (0.4 mg/L) ($n = 108$), PZA (100.0 mg/L) ($n = 118$), and LFX (1.0 mg/L) ($n = 96$). Drug susceptibility testing (DST) for high-dose INH and PZA was performed using the MGIT 960 INH 0.4 and PZA kits (BD BACTEC), respectively, following the manufacturer's instructions, and the susceptibility test for LFX was performed as recommended by the WHO (6).

In addition, MIC values were evaluated for 60 nonduplicate Hr-TB isolates that were randomly selected based on the MGIT results [INH 0.4 resistant ($n = 30$) and INH 0.4 susceptible ($n = 30$)]. However, four isolates (one INH 0.4 resistant and three INH 0.4 susceptible) were excluded because of contamination. Finally, 56 nonduplicate Hr-TB isolates were tested by the broth microdilution method using a custom SEN-SITITRE THAMYCO plate (Thermo Fisher Scientific, Waltham, MA, USA) containing 12 drugs in twofold serial dilutions of RIF, INH, EMB, LFX, moxifloxacin (MFX), bedaquiline (BDQ), linezolid (LZD), clofazimine (CFZ), kanamycin (KAN), amikacin (AMK), ethionamide (ETO), and capreomycin (CAP) (Table S1) following the manufacturer's instructions. Briefly, bacterial cell suspensions (0.5 McFarland standard) were prepared from bacterial colonies and then diluted in 11 mL of 7H9 broth supplemented with oleic albumin dextrose catalase, 100 µL of which (approximately $5 \times 10^5$ CFU/mL) was used for plate inoculation. The plates were sealed and incubated at 37°C for 14–21 days or until sufficient growth was observed in the positive control. Quality controls were performed using *M. tuberculosis* H37Rv (ATCC 27294). MIC results were interpreted, and MTBC isolates were classified into susceptible (S) or resistant (R) categories according to the CLSI breakpoints and epidemiological cut-off values (ECOFF/ECVs) (mg/L) reported by the CRyPTIC (12) (Table S1).

## WGS and data analysis

Genomic DNA extraction from culture materials was performed using the cetyltrimethylammonium bromide–sodium chloride method (13). WGS was performed using the NEBNext Ultra DNA Library Prep Kit for Illumina (New England Biolabs, Ipswich, MA, USA) and a NovaSeq sequencer (Illumina Inc., San Diego, CA, USA) using a $2 \times 150$ paired-end configuration. NovaSeq Control Software, OLB, and GAPipeline-1.6 (Illumina) were used for image analysis and base calling.

Trimmomatic V0.32 (14) and FastQC version 0.11.7 (15) were used to trim and check the quality of the FASTQ files (> Phred score 30). *De novo* assembly was performed with SPAdes (16) and quality-checked with CheckM (17) to validate the completeness (100%) and contamination (0%). Raw FASTQ sequencing data were analyzed using an online tool, TB-Profiler Version 4.4.0, to identify species and strain lineage and for drug susceptibility prediction (98% to 100% frequency of the variant) (18). Snippy software version 4.6.0 (19) was used to identify single-nucleotide polymorphisms (SNPs) between the *M. tuberculosis* genome (GenBank accession number: NC_000962.3) and

the generated core SNP alignment. Phylogenetic analysis of the 56 MTB isolates was performed using MEGA version 11 and a general time-reversible and gamma distribution model (chosen model based on data). The phylogenetic tree was built using 1,000 bootstrap repetitions. The phylogenetic tree was visualized with iTOL.

## Statistical analysis

Data were analyzed using R version 4.2.1. The median and interquartile range (IQR) for age and categorical variables as percentages (%) were calculated using the rstatix package. Positive predictive value (PPV), negative predictive value, and 95% confidence interval (CI) were analyzed using the Yardstick version 1.1.0 package. Graphs were created using the ggplot2 program, version 3.2.1 (20).

## RESULTS

### MTBC clinical isolates and patient populations

From a retrospective study, a total of 144 patients with TB were diagnosed with Hr-TB based on MGIT results, which showed the results of INH resistance and RIF susceptibility. Among the 144 patients with Hr-TB, 63 (43.8%) patients were women and 81 (56.3%) were men, with an age range of 2–88 years (median = 49, IQR = 32.5). Among these, a total of 143 Hr-TB cases with available medical records for review were classified as pulmonary TB ($n$ = 97, 67.8%) and extrapulmonary TB ($n$ = 46, 32.2%) based on history, clinical presentations, and imaging. From the medical record review, the prevalence of Hr-TB in our hospital setting was 11.2% during 2017–2021 (144 Hr-TB patients/1,282 total TB patients with MGIT results). Based on the treatment history, 89.5% (128/143) and 10.5% (15/143) were new and previously treated patients with TB, respectively. The treatment information of one patient was not available. A total of 161 nonduplicate Hr-TB isolates [MTB (151/161, 93.8%); MTBC (10/161, 6.2%)] from 144 Hr-TB patients were recovered from 128 (79.5%) pulmonary specimens (bronchoalveolar lavage fluid, endotracheal aspirate, and sputum) and 33 (20.5%) extrapulmonary specimens (lymph node, tissue biopsy, cerebrospinal fluid, urine, body fluid, gastric content, and pus).

### Phenotypic drug susceptibility patterns of Hr-TB clinical isolates

#### MGIT system

The SIRE MGIT results revealed that 161 nonduplicate Hr-TB isolates (100% R to 0.1 INH and S to RIF) were resistant to STM (69/161, 42.9%) and EMB (3/161, 1.9%). The prospective MGIT study showed that Hr-TB isolates had co-resistance to PZA (24/118, 20.3%) and LFX (2/96, 2.1%). Furthermore, 55.6% (60/108) of the tested Hr-TB isolates were resistant to high-dose INH (0.4), which was classified as high-level INH resistance and exhibited co-resistance to EMB (1/108, 0.9%), PZA (11/108, 10.2%), and LFX (2/96, 2.1%) (Table 1). Two LFX-resistant MTB isolates were recovered from previously treated TB cases, one of which was also resistant to PZA. However, only one (G1-3) of the two isolates was confirmed to be fluoroquinolone (FLQ)-resistant in the WGS analysis but was susceptible to INH (Table S2).

#### Broth microdilution method

The BMD method was performed on 56 Hr-TB isolates (1 MTBC and 55 MTB identified by LPA). However, PZA was not tested using the BMD method due to WHO recommendations (6). The Hr-TB isolates were resistant to INH (53/56, 94.6%), RIF (1/56, 1.8%), ETO (20/56, 35.7%), and KAN (1/56, 1.8%). All 56 Hr-TB isolates were susceptible to EMB, LFX, MFX, BDQ, LZD, CFZ, and AMK. A breakpoint of CAP was not addressed. The categorical agreement of the MGIT and BMD methods was 94.6%, 98.2%, 98.2%, and 96.4% for resistance to low-dose INH, RIF, EMB, and LFX, respectively. A total of 27 and 29 Hr-TB isolates, which were classified as having low- and high-level INH resistance using the MGIT, showed MIC ranges between 0.12 and 2 mg/L (median = 0.25 mg/L) and 0.5 and

**TABLE 1** Drug susceptibility patterns and co-resistance (%) of Hr-TB isolates using the MGIT method[c]

| MGIT | | 0.1 INH MGIT[a] | | 0.4 INH MGIT[b] | | | |
| --- | --- | --- | --- | --- | --- | --- | --- |
| | | No. of isolates (%) | % Co-resistance to 0.1 INH | No. of isolates (%) | | | % Co-resistance to 0.4 INH |
| Drug | Result | R | | S | R | Total | |
| EMB | S | 158 (98.1) | | 47 (43.5) | 59 (54.6) | 106 (98.1) | |
| | R | 3 (1.9) | 1.9 | 1 (0.9) | 1 (0.9) | 2 (1.9) | 0.9 |
| | Total | 161 (100.0) | | 48 (44.4) | 60 (55.6) | 108 (100.0) | |
| PZA | S | 94 (79.7) | | 36 (33.3) | 49 (45.4) | 85 (78.7) | |
| | R | 24 (20.3) | 20.3 | 12 (11.1) | 11 (10.2) | 23 (21.3) | 10.2 |
| | Total | 118 (100.0) | | 48 (44.4) | 60 (55.6) | 108 (100.0) | |
| LFX | S | 94 (97.9) | | 42 (43.8) | 52 (54.2) | 94 (97.9) | |
| | R | 2 (2.1) | 2.1 | 0 | 2 (2.1) | 2 (2.1) | 2.1 |
| | Total | 96 (100.0) | | 42 (43.8) | 54 (56.3) | 96 (100.0) | |

[a]0.1 INH MGIT is the INH critical concentration of 0.1 mg/L.
[b]0.4 INH MGIT is the INH critical concentration of 0.4 mg/L.
[c]No., number; R, resistant; S, susceptible.

>16 mg/L (median = 2 mg/L), respectively. The MIC distributions, ranges, and medians of each drug are shown in Table 2, Fig. 1, and Fig. S1.

## Genotypic characteristics and WGS-based gDST results of Hr-TB clinical isolates

A total of 56 nonduplicate Hr-TB isolates with MIC values were further genetically analyzed using WGS. The median read pairs per isolate were 131,866,600 read pairs (Phred score >30). The read mapping resulted in 470-fold coverage on average. All 56 clinical isolates were identified as MTB and classified into lineage 1 (Indo-Oceanic lineage) (23/56, 41.1%), lineage 2 (East-Asian lineage) (27/56, 48.2%), and lineage 4 (Euro-American lineage) (6/56, 10.7%) (Table S2). Lineage 1 and lineage 2 were the main lineages among Hr-TB isolates.

The prediction of genotypic drug resistance using the TB-Profiler (98% to 100% frequency of the variant) revealed that the 56 MTB clinical isolates exhibited resistance

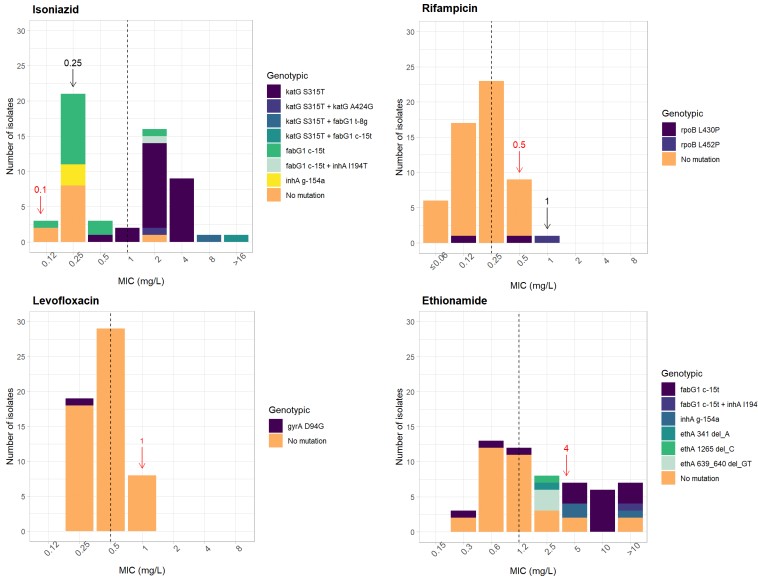

**FIG 1** Distributions of drug resistance-conferring mutations with the corresponding MIC values using the THAMYCO plate. The arrows indicate the ECOFF/ECVs (mg/L) of the CRyPTIC consortium publication (red) and the CLSI breakpoints (mg/L) (black). The dashed lines indicate the median MIC values of Hr-TB isolates (n = 56).

**TABLE 2** Comparison of the results of the broth microdilution method (THAMYCO) and MGIT[b]

| Drug (CC, mg/L) | MGIT Result | No. of isolates | MIC (mg/L) and no. of isolates[a] | | | | | | | | | | | BMD (THAMYCO) Range (median, mg/L) | No. of S isolates | No. of R isolates | % Categorical agreement (95% CI) |
|---|---|---|---|---|---|---|---|---|---|---|---|---|---|---|---|---|---|
| | | | 0.03 | 0.06 | 0.12 | 0.25 | 0.5 | 1 | 2 | 4 | 8 | 16 | >16 | | | | |
| INH (0.1) | R | 56 | 0 | 0 | 3 | 21 | 3 | 2 | 16 | 9 | 1 | 0 | 1 | 0.12–>16 (1) | 3 | 53 | |
| | Total | 56 | 0 | 0 | 3 | 21 | 3 | 2 | 16 | 9 | 1 | 0 | 1 | 0.12–>16 (1) | 3 | 53 | 94.6 (84.2–98.6) |
| | | | ≤0.06 | | 0.12 | 0.25 | 0.5 | 1 | 2 | 4 | 8 | 16 | | | | | |
| RIF (1.0) | S | 56 | 6 | | 17 | 23 | 9 | 1 | 0 | 0 | 0 | 0 | | ≤0.06–1 (0.25) | 55 | 1 | |
| | Total | 56 | 6 | | 17 | 23 | 9 | 1 | 0 | 0 | 0 | 0 | | ≤0.06–1 (0.25) | 55 | 1 | 98.2 (89.2–99.9) |
| | | | | | | 0.25 | 0.5 | 1 | 2 | 4 | 8 | 16 | | | | | |
| EMB (5.0) | S | 55 | | | | 0 | 0 | 5 | 49 | 1 | 0 | 0 | | 1–4 (2) | 55 | 0 | |
| | R | 1 | | | | 0 | 0 | 0 | 1 | 0 | 0 | 0 | | 1 (NA) | 1 | 0 | |
| | Total | 56 | | | | 0 | 0 | 5 | 50 | 1 | 0 | 0 | | 1–4 (2) | 56 | 0 | 98.2 (89.2–99.9) |
| | | | | | 0.12 | 0.25 | 0.5 | 1 | 2 | 4 | 8 | | | | | | |
| LFX (1.0) | S | 54 | | | 0 | 17 | 29 | 8 | 0 | 0 | 0 | | | 0.25–1 (0.5) | 54 | 0 | |
| | R | 2 | | | 0 | 2 | 0 | 0 | 0 | 0 | 0 | | | 2 (NA) | 2 | 0 | |
| | Total | 56 | | | 0 | 19 | 31 | 8 | 0 | 0 | 0 | | | 0.25–1 (0.5) | 56 | 0 | 96.4 (86.6–99.4) |

[a]The vertical bold line divides susceptible and resistant isolates based on the CLSI guidelines and CRyPTIC ECOFF/ECVs.

[b]no., number; CC, critical concentration; R, resistant; S, susceptible.

to RIF (3/56, 5.4%), INH (45/56, 80.4%), STM (14/56, 25.0%), ETO (24/56, 42.9%), and FLQ (1/56, 1.8%). The MTB isolates were classified as Hr-TB (42/56, 75.0%), MDR-TB (3/56, 5.4%), monoresistant to STM (2/56, 3.6%), FLQ- and ETO-resistant TB (1/56, 1.8%), and pansusceptible TB (8/56, 14.3%) based on WGS-based gDST. For DR-TB isolates within lineage 2, all Hr-TB ($n$ = 20), FLQ-R TB ($n$ = 1), and MDR-TB ($n$ = 3) were classified in sublineage 2.2.1. WGS-based gDST results are shown in Table S2.

The categorical agreement between the WGS-based gDST and MGIT methods was high for RIF (94.6%), EMB (98.2%), and LFX (98.2%) but lower for INH (80.4%), PZA (73.2%), and STM (83.9%) (Table S3). Furthermore, the categorical agreement between WGS-based gDST and the BMD method was comparable for RIF (96.4%), FLQ (98.2%), and KAN (98.2%). They were 100% in agreement for EMB, BDQ, LZD, CFZ, and AMK, in which no mutations associated with resistance were detected. However, the agreement between these two methods was lower for INH (82.1%) and ETO (78.6%) (Table S4).

### Isoniazid

INH resistance-conferring mutations were identified in 45/56 (80.4%) Hr-TB isolates. MTB isolates with the *katG* S315T mutation alone (24/45, 53.3%) conferred MIC values of 0.5 to 4 mg/L (median = 2 mg/L), whereas that of MTB isolates with the *inhA* g-154a mutation alone (3/45, 6.7%) was 0.25 mg/L. The isolates with combined mutations of *katG* S315T + *katG* A424G ($n$ = 1), *katG* S315T + *fabG1* t-8g ($n$ = 1), *katG* S315T + *fabG1* c-15t ($n$ = 1), and *fabG1* c-15t + inhA I194T ($n$ = 1) had MIC values of 2, 8, >16, and 2 mg/L, respectively. Moreover, 26/27 (96.3%) MTB isolates with the *katG* S315T mutation were resistant to 0.4 INH by the MGIT method (96.3% PPV). Last, isolates with the *fabG*1 c-15t mutation (*inhA* promoter region) (14/45, 31.1%) conferred a lower MIC ranging from 0.12 to 2 mg/L (median = 0.25 mg/L), of which 13/14 (92.9%) were susceptible to 0.4 INH by the MGIT method. The MTB isolates (11/56, 19.6%) without INH resistance-conferring mutations had MIC values of 0.12–2 mg/L (median = 0.25 mg/L) (Fig. 1; Table S5).

### Rifampicin

Three RIF-R MTB isolates had *rpoB* L430P (2/3, 67.7%) and *rpoB* L452P (1/3, 33.3%) mutations. The MICs were 0.12 (S), 0.5 (S), and 1 (R) mg/L (Fig. 1; Table S2).

### Fluoroquinolone

FLQ resistance was detected in one MTB isolate with a *gyrA* D94G mutation with an MIC value of 0.25 (S) mg/L for LFX and OFX (Fig. 1; Table S2).

### Pyrazinamide

In the MGIT testing, 15 PZA-R MTB isolates were detected. However, the TB-Profiler software identified none of the mutations in the *pncA* gene associated with resistance in the PZA-susceptible (41/56, 73.2%) and PZA-resistant (15/56, 26.8%) MTB isolates. The WGS analysis showed missense mutations in *clpC1*, *rv1258c*, PPE35, and *rv3236c* (Table S6).

### Ethionamide

The common mutations associated with ETO resistance in this study were *fabG1* c-15t (16/24, 66.7%), followed by *ethA* deletion (5/24, 20.8%) and *inhA* g-154a (3/24, 12.5%). An isolate with the *fabG1* c-15t + *inhA* I194T mutation (1/24, 4.16%) had an MIC of >10 mg/L. MTB isolates with only the *fabG1* c-15t mutation had a wide range of MICs (0.03 – >10 mg/L), of which 12/15 (80.0%) were phenotypically resistant. The *ethA* mutations conferred an MIC of 2.5 (S) mg/L. All three isolates with *inhA* g-154a were found to be resistant to ETO by using the BMD method, with MICs of 5, 5, and >10 mg/L (Fig. 1; Table S2).

## DISCUSSION

This study investigated the phenotypic and genotypic susceptibility profiles of Hr-TB isolates at KCMH, Bangkok, Thailand, from 2017 to 2021 and compared the results of Hr-TB diagnostic methods. Hr-TB is the most prevalent DR-TB, with an estimated national prevalence of 9.7% and 21.3% among new and previously treated patients with TB, respectively (21). The current Hr-TB treatment regimen includes RIF, EMB, PZA, and LFX for 6 months (3). Susceptibility testing for INH, RIF, and other drugs is necessary to design an individual treatment regimen with other second-line TB drugs. In the present study, the prevalence of Hr-TB cases was 11.2% from 2017 to 2021 at KCMH, which is a tertiary care hospital; however, most of the cases (89.5%) were newly diagnosed patients with TB. Therefore, an investigation of the transmission of Hr-TB among the study population will be performed in subsequent studies. Furthermore, the low prevalence of previously treated Hr-TB cases in our study could be due to the underreported history of TB treatment in some patients.

SIRE MIGT was the initial pDST performed for MTBC isolates from patients suspected of TB in our hospital setting and categorically classified the MTBC as resistant or susceptible isolates. The BMD method can test up to 12 anti-TB agents simultaneously, and MTB resistance can be quantitatively evaluated as MIC values. However, the CLSI-established breakpoints are limited to RIF, INH, and EMB but not to other drugs due to their limited clinical relevance (7). Recently, the CRyPTIC endorsed ECOFFs/ECVs for various anti-TB drugs (12), which were used to interpret the BMD results in the present study. Our findings demonstrated a strong concordance between the results of the MGIT system and the BMD method in evaluating susceptibility to INH, RIF, EMB, and LFX. In addition, EMB and LFX still have active *in vitro* activity against MTB.

Currently, molecular assays play an important role in the diagnosis of DR-TB. Gene sequencing, specifically WGS, has been implemented for TB surveillance and prediction of drug susceptibility in certain countries (22, 23). WGS results are based on SNP-associated resistance identified using high-/good-performance databases such as the TB-Profiler (18). Therefore, WGS can provide information on mutations that current commercial gDST assays cannot detect. However, for second-line drugs, an additional pDST is necessary to detect resistance that may be missed by the WGS-based gDST due to the lack of sufficient mutation data to support the findings or standardize the bioinformatics pipeline (5, 24).

The phylogenetic analysis of the Hr-TB isolates showed that the most common lineage was lineage 2 (East-Asian lineage), followed by lineage 1 (Indo-Oceanic lineage), and then lineage 4 (Euro-American lineage). Consistent with our results, a previous study has also shown that lineage 2 was the main lineage among DR-TB (MDR, pre-XDR, and XDR-TB) in Thailand (25). For high-level INH resistance, the MGIT results revealed that approximately half of the MTBC isolates (55.6%) were resistant to 0.4 INH. We further analyzed WGS-based gDST and pDST data and found that high- and low-level INH resistance was correlated with the *katG* S315T mutation and mutations in the promoter region of the *inhA-fabG1* operon, respectively, which was consistent with the previous studies (5, 8). Therefore, either pDST or gDST should be performed to avoid the use of INH in isolates with high-level INH resistance.

Furthermore, the categorical agreement between WGS-based gDST and pDST for detection of INH resistance was lower than that for other drugs because WGS-based gDST did not detect mutations associated with INH resistance in 11 Hr-TB isolates, suggesting that there may be new mutations that were not confirmed at the time as being associated with INH resistance due to insufficient evidence (5). This result may also be attributed to repeated subcultures of MTB in fresh medium, which might reduce genetic diversity within the MTB sample due to clone selection (26). Therefore, the discordance of the pDST/gDST results should be further analyzed in future studies.

Furthermore, WGS did not detect mutations associated with PZA resistance (5) in MGIT PZA-R MTB isolates. WGS analysis showed that missense mutations in *clpC1*, *rv1258c*, PPE35, and *rv3236c* are not associated with PZA resistance, and some mutations

are not listed by the WHO (5). At present, MGIT is the only WHO-recommended PZA-pDST; however, this method is associated with a high rate of false-positive resistance due to improper preparation of the inoculum and pH of the culture medium (6, 27, 28). Currently, molecular PZA-DST is the most reliable method recommended by the WHO (6). Therefore, the resistance to PZA revealed using MGIT in our study may be a false-positive result. To confirm our findings, the PZA pDST was retested using the MGIT method, and all 15 PZA-resistant isolates revealed susceptible results. As a result, the initial PZA resistance was likely a false-positive result. However, if this resistance was a true positive, it could be conferred by other mutations (5) or other new mechanisms. In conclusion, clinicians should interpret PZA phenotypic susceptibility with caution, and molecular tests should be performed, in addition to the MGIT, to accurately detect resistance to PZA.

In addition, WGS-based gDST detected three RIF-R MTB isolates that were diagnosed as MDR-TB with *rpoB* L430P ($n = 2$) and *rpoB* L452P ($n = 1$) mutations, which have been previously demonstrated as *rpoB* low-level mutations and could not be detected using the MGIT method (23), consistent with the findings of this study. We also tested a newly proposed RIF critical concentration by the WHO (29), 0.5 mg/L, using the MGIT system. However, no resistance to RIF for these three MTB isolates was detected (unpublished results). Furthermore, the ETO, which shares resistance-associated mutations to INH, *fabG1* c-15t (5), had discordant results between the BMD and WGS-based gDST (Tables S2 and S4). This discrepancy can be caused by the lack of sufficient reproducibility of this drug. Therefore, the pDST is not recommended for ETO by the WHO (6). A discordant result was also observed for FLQ, in which the WGS-based method detected a *gyrA* D94G mutation correlated with high-level MFX resistance in one isolate but was not detected by either pDST. The MICs of FLQ can vary in isolates with the same mutation (30).

The discordant results between the MGIT and BMD methods and between the pDST and gDST were observed for anti-TB drugs. These findings have been well described and recognized in previous studies for several anti-TB drugs and for both the pDST and gDST (24, 31, 32). The discrepancy can directly impact the decision of treatment regimen selection since most laboratories usually have only one validated pDST or gDST method for use. Laboratory personnel and clinicians should be aware that the DST can be discordant with other methods and communicate a possible explanation. The confirmation of DST results by a reference laboratory may be necessary for certain TB cases with complex resistance. Finally, the determination of the risk of drug resistance in patients from a history of prior TB treatment and compliance with anti-TB drugs as well as clinical manifestations are crucial and would assist in treatment regimen selection in this situation or when confirmation cannot be performed.

This study had a few limitations. First, no INH-susceptible MTBC isolate was evaluated for gDST and pDST. In the present study, only 56 Hr-TB clinical isolates were analyzed for broth microdilution and WGS. Moreover, different methods were not performed to reevaluate the discordance of the pDST/gDST results for some drugs. Nevertheless, to the best of our knowledge, this study presented the first susceptibility profiles of the WHO-recommended treatment regimen for Hr-TB isolates from patients in Thailand. Furthermore, the performance of the current pDST used in routine laboratory practices was evaluated and compared to that of the WGS-based gDST.

## Conclusion

We demonstrated that the first-line regimen for Hr-TB still has active activity against the organism *in vitro* using the MGIT and BMD methods. The Hr-TB isolates had a low rate of EMB and LFX resistance, while the high PZA resistance rate was likely a false-positive result. In addition, the co-resistance rate to EMB and LFX was low. Moreover, the Hr-TB isolates were susceptible to other second-line TB drugs, such as BDQ, LZD, and CFZ, which can be used as substitutes in Hr-TB patients who cannot tolerate the adverse effects of the standard regimen. The findings of this study can help clinicians carefully

select and interpret the results of DST for patients with DR-TB, especially those with Hr-TB.

## ACKNOWLEDGMENTS

This study was supported by the Ratchadapiseksompoch Fund, Faculty of Medicine, Chulalongkorn University, grant number RA63/073.

S.R. and S.Pa. conceptualized and designed the study. S.R. and A.P. collected, interpreted, and analyzed the data and drafted the manuscript. S.Pa. and K.F. supervised and guided the study process. S.Pe. and U.S. conducted laboratory experiments. K.W. and K.K. collected and interpreted the clinical data. All authors critically reviewed and revised the manuscript for important intellectual content and agreed to submit the final version for publication.

## AUTHOR AFFILIATIONS

[1]Program in Bioinformatics and Computational Biology, Graduate School, Chulalongkorn University, Bangkok, Thailand

[2]Center of Excellence in Systems Microbiology, Faculty of Medicine, Chulalongkorn University, Bangkok, Thailand

[3]Department of Microbiology, Faculty of Medicine, Chulalongkorn University, Bangkok, Thailand

[4]Division of Infectious Diseases, Department of Medicine, Faculty of Medicine, Chulalongkorn University, Bangkok, Thailand

[5]Department of Microbiology, King Chulalongkorn Memorial Hospital, Bangkok, Thailand

[6]Department of Microbiology, Faculty of Medicine, Khon Kaen University, Khon Kaen, Thailand

[7]Research and Diagnostic Center for Emerging Infectious Diseases, Khon Kaen University, Khon Kaen, Thailand

[8]Center of Excellence in Tuberculosis, Faculty of Medicine, Chulalongkorn University, Bangkok, Thailand

[9]Division of Pulmonary and Critical Care Medicine, Department of Medicine, Faculty of Medicine, Chulalongkorn University, Bangkok, Thailand

[10]Department of Biochemistry, Faculty of Medicine, Chulalongkorn University, Bangkok, Thailand

## AUTHOR ORCIDs

Kanphai Wongjarit http://orcid.org/0000-0002-7460-9674
Kiatichai Faksri http://orcid.org/0000-0001-5022-4182
Sunchai Payungporn http://orcid.org/0000-0003-2668-110X
Suwatchareeporn Rotcheewaphan http://orcid.org/0000-0003-4578-1560

## FUNDING

| Funder | Grant(s) | Author(s) |
| --- | --- | --- |
| CU \| Faculty of Medicine, Chulalongkorn University (MDCU) | RA63/073 | Suwatchareeporn Rotcheewaphan |

## AUTHOR CONTRIBUTIONS

Ajala Prommi, Data curation, Formal analysis, Writing – original draft, Writing – review and editing | Kanphai Wongjarit, Data curation, Formal analysis, Writing – review and editing | Suthidee Petsong, Investigation, Methodology, Writing – review and editing | Ubonwan Somsukpiroh, Investigation, Methodology, Writing – review and editing | Kiatichai Faksri, Supervision, Writing – review and editing | Kamon Kawkitinarong, Data curation, Formal analysis, Writing – review and editing | Sunchai

Payungporn, Conceptualization, Supervision, Writing – review and editing | Suwatchareeporn Rotcheewaphan, Conceptualization, Data curation, Funding acquisition, Writing – original draft, Writing – review and editing

## DATA AVAILABILITY

The WGS data were submitted to the Sequence Read Archive (https://www.ncbi.nlm.nih.gov/sra) with the BioProject accession number PRJNA930660.

## ETHICS APPROVAL

This study was approved by the Institutional Review Board of the Faculty of Medicine, Chulalongkorn University, Bangkok, Thailand (COA.1297/2019).

## ADDITIONAL FILES

The following material is available online.

### Supplemental Material

**Fig. S1, Tables S1 to S6 (Spectrum03462-23-s0001.pdf).** Supplemental figures and tables.

### Open Peer Review

**PEER REVIEW HISTORY (review-history.pdf).** An accounting of the reviewer comments and feedback.

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
