## [Reviewer comments · Microbiology Spectrum]

Microbiology Spectrum

Co-resistance to isoniazid and second-line anti-tuberculosis drugs in isoniazid-resistant tuberculosis at a tertiary care hospital in Thailand

Ajala Prommi, Kanphai Wongjarit, Suthidee Petsong, Ubonwan Somsukpiroh, Kiaticchai Faksri, Kamon Kawkitinarong, Sunchai Payungporn, and Suwatchareeporn Rotcheewaphan

Corresponding Author(s): Suwatchareeporn Rotcheewaphan, Chulalongkorn University

Review Timeline:

Submission Date:	September 25, 2023
Editorial Decision:	November 6, 2023
Revision Received:	December 4, 2023
Accepted:	January 2, 2024

Editor: Sadjia Bekal

Reviewer(s): Disclosure of reviewer identity is with reference to reviewer comments included in decision letter(s). The following individuals involved in review of your submission have agreed to reveal their identity: Asra'a Abdul-Jalil (Reviewer #2); pierre-marie akochy (Reviewer #3)

Transaction Report:

DOI: <https://doi.org/10.1128/spectrum.03462-23>

Re: Spectrum03462-23 (Phenotypic and whole genome sequencing-based drug susceptibility profiles of isoniazid-resistant *Mycobacterium tuberculosis* complex clinical isolates in Thailand)

Dear Dr. Suwatchareeporn Rotcheewaphan:

Thank you for the privilege of reviewing your work. Below you will find my comments, instructions from the Spectrum editorial office, and the reviewer comments.

Revision Guidelines

Sincerely,
Sadjia Bekal
Editor
Microbiology Spectrum

Reviewer #1 (Comments for the Author):

The authors need to select the strains carefully, all Hr-TB strains need to be homogeneously tested by all methods (MGIT and BDM Versus WGS). they need to clarify what is the gold standard method and reference to compare. To improve this paper, I would suggest increasing number of INH-resistant isolates tested using microbroth dilution and WGS (specifically on further isolates already tested by MGIT). Also need to include analysis of susceptible strains in both pDST and gDST analysis for as

control and for comparison. from figure 1 it will be also interesting to include rifampin-reistance strains and some MDR.

Reviewer #2 (Comments for the Author):

Hi dear,

The work is interesting but I didn't find the details of the Accession number (PRJNA930660)on (<https://www.ncbi.nlm.nih.gov/sra>), can you send me the details if the submission is completed?

Best Regard

Reviewer's report

Phenotypic and whole genome sequencing-based drug susceptibility profiles of isoniazid resistant *Mycobacterium tuberculosis* complex clinical isolates in Thailand

Ajala Prommi, Kanphai Wongjarit, Suthidee Petsong, Ubonwan Somsukpiroh, Kamon Kawkitinarong, Sunchai Payungporn, Suwatchareeporn Rotcheewaphan.

This manuscript submitted by Prommi A. et al. describes drug susceptibility profiles of INH-resistant Tuberculosis (Hr-TB) and investigates mutations associated with anti-tuberculous drugs in Hr-TB clinical isolates from a hospital laboratory in Thailand. Phenotypic and genotypic drug susceptibility testing (pDST and gDST) was retrospectively and prospectively investigated using two phenotypic methods and one whole genome sequencing-based genomic method.

Major Comments

- 1- There is a design issue in this study. First, the authors justify the importance of their study in section "Importance" Lines 45-51. They state the susceptibility testing of Hr-TB clinical isolates is critical in the selection of the appropriate treatment regimen to prevent failure of treatment and the development of multidrug-resistant TB (MDR-TB). But they did not support this in the manuscript.
- 2- Also they claimed their study highlights the susceptibility profiles of Hr-TB clinical isolates from Thailand (not true, this a single hospital study) and the performance and importance of using pDST and gDST for accurate and comprehensive result interpretation. From the study finding it is not clear how the interpretation was made in case of discrepancies between pDST and gDST and what is the gold standard method used in this study. The authors should clarify what is the phenotypic gold standard in case of discrepancy between MGIT and BDM, and what is reported or advised to clinician / reader to report in case of discrepancy between pDST and gDST. How these different scenarios were reported and impacted the TB patient treatment ?
- 3- I understand that an appreciated work is conducted retrospectively and prospectively on specific and interesting topics (Hr-TB), The authors diluted the importance of Hr-TB by describing more data on the total cases studied during the five years period. It would be more attracting to focus on Hr-TB that are characterized by both MGIT and BDM and WGS.

To improve this paper, I would suggest increasing number of INH-resistant isolates tested using microbroth dilution and WGS (specifically on further isolates already tested by MGIT). Also need to include analysis of susceptible strains in both pDST and gDST analysis for comparison. There are many sentences that need to be clarified or simplified (common theme throughout is how did 56 INH-resistant isolates appear? Where are these derived from? Why were they selected as a subset? Some drugs have DST data, some have AMR predictions...so confusing).

Related to the above (Line 198) – why only 56 of 161 Hr-TB isolates were selected? What were the selection criteria to only testing these 56 microbroth method? In Lines 297-298, they discuss this as a limitation, but never clarify why it was limited to 56?

- 4- This study needs to include more isolates derived from sensitive TB, and other Resistant TB available in their collection. It would also be nice to know mutation data for phenotypic INH sensitive strains (non associated, low confidence mutations or discrepant) and RR-TB and MDR-isolates for comparison. Looking at only Hr strains is selective and adds bias to the study. It is well known that many mutations are also found in phenotypic susceptible isolates (See CRYPTIC consortium and WHO mutation catalog).

General comments

In the Abstract section they mention that the prevalence of Hr-TB cases was 11,2% based on 144 Hr-TB patients out of 1,282 total cases in this study (patients population section and table 1). How they determine this prevalence? It is clear this is a period prevalence from 2017-2012 at KCMH. But is it a hospital lab-based calculation? city or region reference lab? is this the only lab receiving the MTB isolates from the region? it will be important to calculate this prevalence over time and describe the space / population number comparing to the national prevalence (may be put it as supplemental data along with other Epi information).

At the local hospital, assuming this prevalence is stable, this is higher than the global Hr-TB prevalence estimated by WHO at 8%. The authors should give more information on the treatment course, outcome and possibly compare it to susceptible TB and MDR/RR-TB (Multidrug resistant TB (31 cases) /rifampin-resistant TB (8 cases). As they mention clinical data are available, one particular analysis that the author should do is to look at Hr-TB treatment output in both associated mutation types (low or high level Isoniazid resistance)

Minor comments

- 1- Study aim sentence may be specific, descriptive and strengthened
- 2- MDR definition needs to be reworded
- 3- Lines 71-72 need to be reworded: Low confidence mutation and low level INH resistance are not essentially interchangeable.
- 4- WGS method is incomplete. QC is not listed. Why was phylogenetic tree needed? Then it is not discussed? Figure S2 is missing distance value and it is not clear if it adds a value to this study in terms of relatedness, Hr-TB transmission, or clustering.
- 5- WGS-based identification is briefly mentioned using TB profiler, the Microdilution was performed on 56 Hr-TB isolates (1 MTBC, 55 MTB). But when looking to WGS, all 56 isolates were identified as MTB. Is the one isolate identified as MTBC also identified as MTB by TB profiler? This needs to be discussed. And more detail on identification methods (probe and WGS pipeline) needs to be clarified and discussed. Differentiation of MTBC species by WGS is widely accomplished by identifying the number of species-specific reads in each sample using different approaches, which interrogate significantly more of the genome rather than one specific short sequence used in probe assay.
- 6- Table 1 DST has several issues: no rifampin data, why the number of isolates for other drugs more than 5?...what are these other isolates?, more drugs are mentioned in the text than in the table.
- 7- The authors discuss first and second-line antimicrobials, but really only focus on INH, RIF, LFX, and ETO....so why even bring up the others? For example, this is not clear why they give some details on Streptomycin and Ethionamid, not clear if Streptomycin is still tested and used for treatment in Thailand and why mutation associated with SM resistance are provided ?
- 5- Line 271: PZA identification, please reword to PZA susceptibility testing.
In the MGIT testing, 15 PZA-R MTB were detected. However, the TB-Profiler identified none of the mutations in the *pncA* gene commonly associated with PZA resistance. MTB (15/56). This is important and I am curious to know if it is unique to the region or it is clonal spread of some mutations. Absence of *pncA* mutations in this collection is

intriguing and need to be discussed. The WGS analysis showed missense mutations in *clpC1*, *rv1258c*, *PPE35*, and *rv3236c*. Discussion of these mutations would be helpful.

- 8- Lines 97-100 – description of the unique 144 patients with Hr-TB isolates derived from all data is confusing
- 9- Line 109 – medium written twice
- 10- Lines 170-172 – the number of RIF resistant + INH resistant + MDR =183, but state that 1,111 or 1,282 are susceptible. There should only be a total of 171 patients with a resistant isolate. These numbers need to be clarified.
- 11- Line 186 – nonduplicatae (typo)
- 12- Line 228 – ET0 – should be ETO , you used a zero as a letter O
- 13- Line 309 – typo “catagorical” categorical
- 14- Lines 309-316 – I like that the authors discuss potential reasons for discordant INH results between gDST and pDST, bringing up both unknown mutations and the potential impact of repeated subculturing without antibiotic pressure. They recognize this is needing further research.
- 15- Line 327 – typo “WSG-based” - WGS-based

General Comments: The authors presented a very interesting work on “
Phenotypic and whole genome sequencing-based drug susceptibility profiles of isoniazid-resistant *Mycobacterium tuberculosis* complex clinical isolates in Thailand.”

The results presented are convincing and the conclusions drawn are supported by the data presented. The study shows for the first-time performances of two routine clinical microbiology laboratory pDST methods and the WGS approach over isolates Hr-TB in Thailand.

Specific Comments:

I have no major criticisms to offer but a specific comment that will intend to improve impact.

1. Given 11 discrepancies between the WGS-based gDST and MGIT method for INH, could Authors evaluate isolates with another pipeline such as Mykrobe predictor.
2. Replace MGIT by BMD in legends of Table S4

Wishing you all the best,

pm

Response to Reviewers

Co-resistance to isoniazid and second-line anti-tuberculosis drugs in isoniazid-resistant tuberculosis at a tertiary care hospital in Thailand

Ajala Prommi, Kanphai Wongjarit, Suthidee Petsong, Ubonwan Somsukpiroh, Kiaticchai Faksri, Kamon Kawkitinarong, Sunchai Payungporn, Suwatchareeporn Rotcheewaphan

Reviewer #1

Major comments

1. There is a design issue in this study. First, the authors justify the importance of their study in section "Importance" Lines 45-51. They state the susceptibility testing of Hr-TB clinical isolates is critical in the selection of the appropriate treatment regimen to prevent failure of treatment and the development of multidrug-resistant TB (MDR-TB). But they did not support this in the manuscript.

Response to the reviewer

First, we appreciate the comments from reviewer 1 that allowed us to have an opportunity to carefully revise the manuscript.

The importance (Line 52-59) and discussion (Line 369-376) sections were revised regarding the comment. The susceptibility profiles of Hr-TB clinical isolates tested by phenotypic and genotypic methods are important in the selection of treatment regimens. A 6-month course of rifampicin, ethambutol, pyrazinamide, and levofloxacin is a preferred regimen for Hr-TB based on the current WHO recommendation (1, 2). However, the regimen needs to be designed individually for patients when additional resistance to pyrazinamide and levofloxacin is present.

2. Also they claimed their study highlights the susceptibility profiles of Hr-TB clinical isolates from Thailand (not true, this a single hospital study) and the performance and importance of using pDST and gDST for accurate and comprehensive result interpretation. From the study finding it is not clear how the interpretation was made in case of discrepancies between pDST and gDST and what is the gold standard method used in this study. The authors should clarify what is the phenotypic gold standard in case of discrepancy

between MGIT and BDM, and what is reported or advised to clinician/reader to report in case of discrepancy between pDST and gDST. How these different scenarios were reported and impacted the TB patient treatment?

Response to reviewer

The title was revised to “Co-resistance to isoniazid and second-line anti-tuberculosis drugs in isoniazid-resistant tuberculosis at a tertiary care hospital in Thailand”.

Regarding the gold standard method, currently, there are several new phenotypic and genotypic DST methods developed and available for testing drug-resistant TB. The advantages and disadvantages of each method have been demonstrated. The discordant results between the DST methods have been well described and recognized in several previous studies for several anti-TB drugs and for both the pDST and gDST (3-6) due to the test performance and nature of some anti-TB drugs. It is difficult to determine which method is the gold standard for all anti-TB agents in the real clinical setting. The WHO states that culture-based phenotypic DST methods are currently the gold standard for drug resistance detection (7) and recommends several phenotypic methods to test drug-resistant TB, including agar proportion, MGIT, and broth microdilution. The MGIT uses a single critical concentration to test the isolate and provides results as a category of S, I, and R, while the BMD provides quantitative MIC results. The WHO recommends performing the pDST because it plays an important role in identifying resistance that cannot be detected by the gDST and in supporting the interpretation of gDST results.

In our opinion, when there is a discrepancy between these two methods, the type of anti-TB agent tested should be considered because of the difference in test performance for different drugs. The agreement between the MGIT and BMD was good for INH (94.6%), RIF (98.2%), EMB (98.2%), and LFX (96.4%), which was consistent with a previous study (8). However, BMD can detect low-level resistance to rifampicin, which cannot be detected by the MGIT method. For drugs without interpretative breakpoints, the MGIT can provide results as S, I, and R when tested as WHO- or CLSI-recommended critical concentrations (7, 9). In addition, the gDST should be performed for PZA, EMB, and ETO because of the unreliability of the pDST. Similarly, laboratories and clinicians should be aware of the performance of the methods used and the presence of discrepancies between the pDST and gDST.

The WGS-based DST has promising performance and is reliable for predicting susceptibility to first-line drugs compared to the pDST method. However, complete mutation data for the detection of second-line drug resistance are still lacking. More evaluation studies need to

be conducted (10). In a real clinical setting, our laboratory independently reports results from each assay. The result confirmation by an alternative method may be performed in some cases if the discrepancy is present for an important drug in a treatment regimen such as isoniazid, rifampicin, or fluoroquinolones, which are critical for the treatment outcome. In addition, these discrepant results should be discussed between clinicians and microbiologists for relevant clinical decision making.

The advantages and disadvantages of each diagnostic assay have been added to the discussion section (Lines 292 -294, 304-306). The explanation is discussed in Lines 325-360 in the discussion section.

3. I understand that an appreciated work is conducted retrospectively and prospectively on specific and interesting topics (Hr-TB), The authors diluted the importance of Hr-TB by describing more data on the total cases studied during the five years period. It would be more attracting to focus on Hr-TB that are characterized by both MGIT and BDM and WGS. To improve this paper, I would suggest increasing number of INH-resistant isolates tested using microbroth dilution and WGS (specifically on further isolates already tested by MGIT). Also need to include analysis of susceptible strains in both pDST and gDST analysis for comparison. There are many sentences that need to be clarified or simplified (common theme throughout is how did 56 INH-resistant isolates appear? Where are these derived from? Why were they selected as a subset? Some drugs have DST data, some have AMR predictions...so confusing). Related to the above (Line 198) – why only 56 of 161 Hr-TB isolates were selected? What were the selection criteria to only testing these 56 microbroth method? In Lines 297-298, they discuss this as a limitation, but never clarify why it was limited to 56?

Response to reviewer

In this work, we report the drug susceptibility patterns of isoniazid-resistant *M. tuberculosis* isolated from patients at our hospital in Thailand using phenotypic and genotypic methods. However, clinical outcomes, response to treatment, and other types of drug resistance, such as MDR-TB, are beyond the scope of this project. Unfortunately, we cannot conduct any further experiments (MIC and WGS) for additional susceptible and resistant MTB isolates since this project was closed. Therefore, access to a BSL3 facility is limited. In addition, the customized MIC plate, not commercial, containing TB drugs that are used in the current regimen for drug-resistant TB recommended by the WHO, such as bedaquiline and clofazimine, is no longer available.

The 56 isolates in this study were selected based on the MGIT results. Initially, there were 60 Hr-TB isolates, including 30 INH 0.4-resistant and 30 INH 0.4-susceptible isolates. However, the results of 4 isolates (one INH 0.4 R and three INH 0.4 S) were excluded from the study due to bacterial contamination and mixed organisms.

The manuscript was revised according to the reviewer's recommendation. The data of total cases and other types of drug-resistant TB that diluted the importance of Hr-TB were removed. The Methods (Lines 101-107) and Results (Lines 178-180) sections were revised. We decided to remove Figure 1 because it may lead to misinformation to the readers. Some patients may have more than one isolate with different drug susceptibility patterns. For example, a patient may have both MDR-TB and Hr-TB or susceptible and Hr-TB isolated from different clinical specimens. Therefore, these patients were included in both the MDR-TB and Hr-TB groups or the susceptible and Hr-TB groups.

4. This study needs to include more isolates derived from sensitive TB, and other Resistant TB available in their collection. It would also be nice to know mutation data for phenotypic INH sensitive strains (non associated, low confidence mutations or discrepant) and RR-TB and MDR-isolates for comparison. Looking at only Hr strains is selective and adds bias to the study. It is well known that many mutations are also found in phenotypic susceptible isolates (See CRYPTIC consortium and WHO mutation catalog).

Response to reviewer

We agree with the reviewer that susceptible INH isolates should be tested for MIC and WGS to improve the understanding of the detected mutations, whether they are associated with resistance, associated with resistance interim, uncertain significance, not associated with resistance interim, or not associated with resistance classification (10). However, the main objective of this study was to investigate the phenotypic and genotypic resistance patterns and rates of drugs in the Hr-TB treatment regimen at our hospital. Therefore, the susceptible isolates were not initially recruited into the study. In addition, this study focused on Hr-TB isolates, which were analyzed further for additional drugs if resistance to second-line drugs such as fluoroquinolone is suspected from clinical manifestations or the risk of drug resistance in patients relevant to routine clinical practice. Furthermore, WGS analysis was performed to investigate mutations present in them. Other types of drug resistance (RR-TB and MDR-TB) are beyond the scope of this project.

General comments

In the Abstract section they mention that the prevalence of Hr-TB cases was 11,2% based on 144 Hr-TB patients out of 1,282 total cases in this study (patients population section and table 1). How they determine this prevalence? It is clear this is a period prevalence from 2017-2012 at KCMH. However, But is it a hospital lab-based calculation? city or region reference lab? is this the only lab receiving the MTB isolates from the region? it will be important to calculate this prevalence over time and describe the space / population number comparing to the national prevalence (may be put it as supplemental data along with other Epi information).

At the local hospital, assuming this prevalence is stable, this is higher than the global Hr-TB prevalence estimated by WHO at 8%. The authors should give more information on the treatment course, outcome and possibly compare it to susceptible TB and MDR/RR-TB (Multidrug resistant TB (31 cases) /rifampin-resistant TB (8 cases). As they mention clinical data are available, one particular analysis that the author should do is to look at Hr-TB treatment output in both associated mutation types (low or high level Isoniazid resistance)

Response to reviewer

The prevalence reported in this study was calculated from a single hospital during the time period of 2017 to 2021. The KCMH is a tertiary care hospital with a TB referral center that may lead to a high prevalence of Hr-TB. Thailand was previously listed as a 14 high TB burden and drug-resistant TB country. As a result, the prevalence of Hr-TB could be higher than that in the global report. However, Thailand is listed as a 30 high TB burden country and is no longer listed in high-burden countries for drug-resistant TB since 2020.

The prevalence of Hr-TB in Thailand was reported in a previous study (Line 280-281) as 9.7% of Hr-TB in new TB patients, which is higher than the WHO report. In addition, a report from another hospital in Bangkok revealed a high prevalence rate of Hr-TB (11.8%) in new patients with positive AFB stain, which was close to the number in our hospital (11). A minor modification was added to Lines 284-285 in the discussion section.

Minor comments

1. Study aim sentence may be specific, descriptive and strengthened

Response: The sentence has been revised in the introduction section (Lines 91-95).

2. MDR definition needs to be reworded

Response: The MDR definition has been revised (Lines 65-67).

3. Lines 71-72 need to be reworded: Low confidence mutation and low level INH resistance are not essentially interchangeable.

Response: The sentence has been revised (Lines 78-82).

4. WGS method is incomplete. QC is not listed. Why was phylogenetic tree needed? Then it is not discussed? Figure S2 is missing distance value and it's not clear if it add a value to this study in term of relatedness, Hr-TB transmission, or clustering.

Response: The WGS method has been revised (Lines 156-158). We agree with the comment of the reviewer that this figure is beyond the scope of the study aim. Thus, we decided to remove Figure S2 phylogenetic tree from the manuscript.

5. WGS-based identification is briefly mentioned using TB profiler, the Microdilution was performed on 56 Hr-TB isolates (1 MTBC, 55 MTB). But when looking to WGS, all 56 isolates were identified as MTB. Is the one isolate identified as MTBC was also identified as MTB by TB profiler? This needs to be discussed. And more detail on identification methods (probe and WGS pipeline) needs to be clarified and discussed. Differentiation of MTBC species by WGS is widely accomplished by identifying the number of species-specific reads in each sample using different approaches, which interrogate significantly more of the genome rather than one specific short sequence used in probe assay.

Response: We apologize for the unclear information. At the time that we performed a broth microdilution assay, the identification of MTBC species was achieved by LPA using Mycobacterium CM VER 2.0 or GenoType MTBC VER 1. X, which can identify the organism to MTBC and MTB species levels, respectively. All isolates were further analyzed by WGS, and all of them were identified as MTB (Lines 220-221). The Methods section has been revised (Lines 113-116).

6. Table 1 DST has several issues: no rifampin data, why the number of isolates for other drugs more than 5?...what are these other isolates?, more drugs are mentioned in the text than in the table.

Response: Table 1 demonstrates the results and coresistance of drugs tested by MGIT (SIRE, levofloxacin, INH 0.4), which is the initial method that most laboratories perform. Then, the isolates were randomly selected based on MGIT results (INH 0.4 susceptible or resistant) for further MIC and WGS analysis. However, rifampicin was not included in this table because all Hr-TB isolates were 100% susceptible to rifampicin by the MGIT method. Therefore, coresistance between rifampicin and isoniazid was not detected. The number of isolates tested by each drug was different because some clinical isolates were excluded from prospective pDST due to bacterial contamination and no growth after subculture, which is a limitation of this study (Lines 126-127). In the text, the MGIT results of SIRE, INH 0.4, and LFX were reported. Additional drugs were reported for a broth microdilution (12 drugs) and WGS sequencing analysis and described in the text.

7. The authors discuss first and second-line antimicrobials, but really only focus on INH, RIF, LFX, and ETO....so why even bring up the others? For example, this is not clear why they give some details on Streptomycin and Ethionamid, not clear if Streptomycin is still tested and used for treatment in Thailand and why mutation associated with SM resistance are provided?

Response: We would like to focus on the Hr-TB treatment regimen. Thus, drugs in the first-line regimen were discussed more often than others. However, ETO was mentioned because it has cross-genetic resistance to INH (such as *fabG1* c-15t), and resistance to this drug was observed by a broth microdilution method.

Currently, streptomycin is classified as a second-line drug used in MDR-TB patients who cannot tolerate amikacin. We agree with the reviewer that this drug may not play a role in Hr-TB treatment. Therefore, the streptomycin MGIT results were removed to make the manuscript concise with only relevant information. The streptomycin MGIT results are reported in the Results section (Lines 196), and the mutations associated with streptomycin and ethionamide are shown in Table S2 to reveal the comprehensive results of drug resistance present in Hr-TB isolates from our institute. In addition, the abstract was revised by removing the streptomycin and ethionamide results (Lines 41-42).

8. Line 271: PZA identification, please reword to PZA susceptibility testing. In the MGIT testing, 15 PZA-R MTB were detected. However, the TB-Profler identified none of the mutations in the *pncA* gene commonly associated with PZA resistance. MTB (15/56). This is important and I am curious to know if it's unique to the region or it is clonal spread of some mutations. Absence of *pncA* mutations in this collection is intriguing and need to be discussed. The WGS analysis showed missense mutations in *clpC1*, *rv1258c*, *PPE35*, and *rv3236c*. Discussion of these mutations would be helpful.

Response: PZA identification was reworded to Pyrazinamide (Line 262), consistent with other drugs.

We are concerned about this issue and agree with the reviewer. Therefore, we repeated the MGIT for PZA while we are waiting for the reviewer's response. The results of 15 PZA MGIT were susceptible in a repeat set. This false-resistance situation is well described and reported in several publications as well as WHO recommendations. The molecular test should be confirmed when the MGIT PZA is resistant. However, based on our experience, most laboratories do not repeat DST tests in the real clinical setting. Therefore, the interpretation of the PZA MGIT result should be done with caution. Other information, especially clinical information, should be considered. The discussion section was revised and included this explanation (Lines 325-337).

However, Table 1 still shows the original PZA results, and a discussion about this issue was added to the discussion section.

9. Lines 97-100 – description of the unique 144 patients with Hr-TB isolates derived from all data is confusing

Response: The Methods section has been revised (Lines 101-107).

10. Line 109 – medium written twice

Response: Thank you and it was corrected.

11. Lines 170-172 – the number of RIF resistant + INH resistant + MDR =183, but state that 1,111 or 1,282 are susceptible. There should only be a total of 171 patients with a resistant isolate. These numbers need to be clarified.

Response: Thank you so much. This information was removed as explained in response to reviewer 3.

12. Line 186 – nonduplicatae (typo)

Response: Thank you. The word has been corrected.

13. Line 228 – ET0 – should be ETO, you used a zero as a letter O

Response: Thank you. The word was corrected.

14. Line 309 – typo “catagorical” categorical

Response: Thank you. The word was corrected.

15. Lines 309-316 – I like that the authors discuss potential reasons for discordant INH results between gDST and pDST, bringing up both unknown mutations and the potential impact of repeated subculturing without antibiotic pressure. They recognize this is needing further research.

Response: Thank you.

16. Line 327 – typo “WSG-based” - WGS-based

Response: Thank you. The word was corrected.

Reviewer #2

The work is interesting but I didn't find the details of the Accession number (PRJNA930660) on (<https://www.ncbi.nlm.nih.gov/sra>), can you send me the details if the submission is completed?

Response to reviewer: Thank you very much. The data will be released to the public on NCBI when the manuscript is accepted for publication.

References

1. WHO. WHO treatment guidelines for isoniazid-resistant tuberculosis: Supplement to the WHO treatment guidelines for drug-resistant tuberculosis. Geneva: World Health Organization; 2018. Licence: CC BY-NC-SA 3.0 IGO.

2. WHO consolidated guidelines on tuberculosis. Module 4: treatment - drug-resistant tuberculosis treatment, 2022 update. Geneva: World Health Organization; 2022. Licence: CC BY-NC-SA 3.0 IGO.
3. Puyén ZM, Santos-Lázaro D, Vigo AN, Coronel J, Alarcón MJ, Cotrina VV, Moore DAJ. Evaluation of the broth microdilution plate methodology for susceptibility testing of *Mycobacterium tuberculosis* in Peru. BMC Infectious Diseases. 2022;22(1):705.
4. Banu S, Rahman SMM, Khan MSR, Ferdous SS, Ahmed S, Gratz J, et al. Discordance across several methods for drug susceptibility testing of drug-resistant *Mycobacterium tuberculosis* isolates in a single laboratory. Journal of Clinical Microbiology. 2014;52(1):156-63.
5. Gygli SM, Keller PM, Ballif M, Blöchli N, Hömke R, Reinhard M, et al. Whole-genome sequencing for drug resistance profile prediction in *Mycobacterium tuberculosis*. Antimicrobial Agents and Chemotherapy. 2019;63(4):10.1128/aac.02175-18.
6. Zhang Z, Wang Y, Pang Y, Liu C. Comparison of different drug susceptibility test methods to detect rifampin heteroresistance in *Mycobacterium tuberculosis*. Antimicrob Agents Chemother. 2014;58(9):5632-5.
7. WHO. Technical manual for drug susceptibility testing of medicines used in the treatment of tuberculosis. Geneva: World Health Organization; 2018. Licence: CC BY-NC-SA 3.0 IGO.
8. Consortium TC. Epidemiological cut-off values for a 96-well broth microdilution plate for high-throughput research antibiotic susceptibility testing of *M. tuberculosis*. European Respiratory Journal. 2022:2200239.
9. CLSI. Performance Standards for Susceptibility Testing of Mycobacteria, *Nocardia spp.*, and Other Aerobic Actinomycetes. 2nd edition. CLSI supplement M24S. Clinical and Laboratory Standards Institute; 2023.
10. WHO. Catalogue of mutations in *Mycobacterium tuberculosis* complex and their association with drug resistance. Geneva: World Health Organization; 2021. Licence: CC BY-NC-SA 3.0 IGO.

11. Kateruttanakul P, Unsematham S. Drug resistance among new smear-positive pulmonary tuberculosis cases in Thailand. *Int J Tuberc Lung Dis.* 2013;17(6):814-7.

Re: Spectrum03462-23R1 (Co-resistance to isoniazid and second-line anti-tuberculosis drugs in isoniazid-resistant tuberculosis at a tertiary care hospital in Thailand)

Dear Dr. Suwatchareeporn Rotcheewaphan:

Your manuscript has been accepted, and I am forwarding it to the ASM production staff for publication. Your paper will first be checked to make sure all elements meet the technical requirements. ASM staff will contact you if anything needs to be revised before copyediting and production can begin. Otherwise, you will be notified when your proofs are ready to be viewed.

Sincerely,
Sadjia Bekal
Editor
Microbiology Spectrum

Reviewer #1 (Comments for the Author):

The revised manuscript is greatly improved, and highlight the importance of HR-TB, and how WGS can resolve or add more information to resistant TB management.

Thank you for your effort and quick reply to both major and minor comments.

Co-resistance to isoniazid and second-line anti-tuberculosis drugs in isoniazid-resistant tuberculosis at a tertiary care hospital in Thailand

Ajala Prommi, Kanphai Wongjarit, Suthidee Petsong, Ubonwan Somsukpiroh, Kiatichai Faksri, Kamon Kawkitinarong, Sunchai Payungporn, Suwatchareeporn Rotcheewaphan

Reviewer #1

The revised manuscript is greatly improved, and highlight the importance of HR-TB, and how WGS can resolve or add more information to resistant TB management. Thank you for your effort and quick reply to both major and minor comments.

Minor comments

1. Line 271: PZA identification, please reword to PZA susceptibility testing. In the MGIT testing, 15 PZA-R MTB were detected. However, the TB-Profiler identified none of the mutations in the *pncA* gene commonly associated with PZA resistance. MTB (15/56). This is important and I am curious to know if it's unique to the region or it is clonal spread of some mutations. Absence of *pncA* mutations in this collection is intriguing and need to be discussed. The WGS analysis showed missense mutations in *clpC1*, *rv1258c*, *PPE35*, and *rv3236c*. Discussion of these mutations would be helpful.

Response: PZA identification was reworded to Pyrazinamide (Line 262), consistent with other drugs.

We are concerned about this issue and agree with the reviewer. Therefore, we repeated the MGIT for PZA while we are waiting for the reviewer's response. The results of 15 PZA MGIT were susceptible in a repeat set. This false-resistance situation is well described and reported in several publications as well as WHO recommendations. The molecular test should be confirmed when the MGIT PZA is resistant. However, based on our experience, most laboratories do not repeat DST tests in the real clinical setting. Therefore, the interpretation of the PZA MGIT result should be done with caution. Other information, especially clinical information, should be considered. The discussion section was revised and included this explanation (Lines 325-337).

However, Table 1 still shows the original PZA results, and a discussion about this issue was added to the discussion section.

Reviewer's response: Thank you for repeating PZA testing, this is what most mycobacteriology labs should do knowing PZA is difficult to test in acidic media. I think it s worthy to mention it in your result section to avoid confusion, I recommend to add it in Result section , line 266 and mention that PZA susceptibility was repeated using MGIT method and were found all susceptible to PZA.